# Association between atrial fibrillation and risk of end-stage renal disease among adults with diabetes mellitus

Yu-Kang Chang[1,2,3], Hueng-Chuen Fan[4,5], Chi-Chien Lin[2,6], Yuan-Hung Wang[7], Wan-Ni Tsai[8☯], Paik-Seong Lim[9☯]*

1 Department of Medical Research, Tungs' Taichung MetroHarbor Hospital, Taichung, Taiwan, 2 Department of Post-Baccalaureate Medicine, College of Medicine, National Chung Hsing University, Taichung, Taiwan, 3 Department of Nursing, Jenteh Junior College of Medicine, Nursing and Management, Miaoli, Taiwan, 4 Department of Pediatrics, Tungs' Taichung MetroHarbor Hospital, Taichung, Taiwan, 5 Department of Rehabilitation, Jenteh Junior College of Medicine, Nursing and Management, Miaoli, Taiwan, 6 Institute of Biomedical Science, iEGG and Animal Biotechnology Center, National Chung-Hsing University, Taichung, Taiwan, 7 Department of Medical Research, Shuang Ho Hospital, Taipei Medical University, New City, Taiwan, 8 Department of Endocrinology and Metabolism, Tungs' Taichung MetroHarbor Hospital, Taichung, Taiwan, 9 Division of Renal Medicine, Tungs' Taichung MetroHarbor Hospital, Taichung, Taiwan

☯ These authors contributed equally to this work.
* jamespslim@gmail.com.

**Data Availability Statement:** Third party NHRID data for this study are stored and maintained by NHRI. We ask that readers refer to the NHRI website (http://nhird.nhri.org.tw/en/Data_Subsets.

## Abstract

Diabetes mellitus (DM) is an important risk factor in patients with end-stage renal disease (ESRD). DM is associated with the development of cardiovascular diseases, such as atrial fibrillation (AF), due to poor glycemic control. However, few studies have focused on the risk of developing ESRD among DM patients with and without AF. This study evaluated ESRD risk among DM patients with and without AF in Taiwan. Data were retrieved from one million patients randomly sampled from Taiwan's National Health Insurance Research Database, including 6,105 DM patients with AF propensity score–matched with 6,105 DM patients without AF. Both groups were followed until death, any dialysis treatment, or December 31, 2013, whichever occurred first. AF was diagnosed by a qualified physician according to the International Classification of Diseases, 9th Revision, Clinical Modification (ICD-9-CM), using the diagnostic code 427.31. Patients aged <20 years or diagnosed with ESRD before the index date were excluded. A Cox proportional hazard regression model was used to calculate the relative ESRD risk. Among DM patients, those with AF have more comorbidities than those without AF. We also found a 1.18-fold (95% confidence interval [CI]: 1.01–1.46) increase in ESRD risk among patients with AF compared with those without AF. In addition, DM patients with hypertension, chronic kidney disease (CKD), or higher Charlson Comorbidity Index scores also have significantly increased ESRD risks than those without these complications. A 1.39-fold (95% CI: 1.04–1.86) increase in risk was observed for patients with AF among the non-CKD group. Our findings suggest that patients with DM should be closely monitored for irregular or rapid heart rates.

html) for more detailed information on these datasets. NHIRD data are available upon request from NHRI (nhird@nhri.edu.tw), for researchers who meet the criteria for access to confidential data. The authors confirm that others would be able to access these third party data from NHRI in the same manner as the authors. The authors also confirm that they did not have any special access privileges that others would not have.

**Competing interests:** The authors have declared that no competing interests exist.

## Introduction

Diabetes mellitus (DM) is a major public health priority worldwide, and the global prevalence of DM is projected to increase from 463 million (9.3%) in 2019 to 700 million (10.9%) in 2045 [1]. Patients with DM have high risks of cardiovascular morbidity and mortality [2], and DM is a leading cause of end-stage renal disease (ESRD), as approximately 40% of all patients with ESRD also have DM [3]. According to the United States Renal Data System annual report, Taiwan has the highest ESRD incidence and prevalence rates worldwide [4], leading to a heavy financial burden on Taiwan's health care systems. Thus, preventive strategies that reduce the risks of developing ESRD should be implemented among patients with DM.

Among patients with ESRD, cardiovascular disease is considered a leading cause of mortality [5], accounting for 48% of total deaths [6]. The most common cardiac arrhythmia is atrial fibrillation (AF), characterized by rapid and irregular atrial activation and associated with increased risks of poor outcomes, such as stroke, heart failure, and mortality [7]. AF and ESRD share risk factors, such as age, hypertension, obesity, DM, vascular heart disease, and heart failure [8], and patients with ESRD on dialysis have a higher AF prevalence (13%–27%) than the general population (approximately 1%), suggesting that uremia may be associated with AF [9–11]. In addition, previous studies have shown that AF is associated with increased risks of stroke and mortality among patients with ESRD on dialysis [12, 13]. The risk of developing AF is even higher among patients with ESRD on dialysis who have one or more of the following risk factors: advanced age, hypertension, heart failure, coronary artery disease (CAD), peripheral vascular disease, and chronic obstructive pulmonary disease (COPD) [14, 15].

AF is associated with poor outcomes among patients with ESRD [16], and the incidence of AF was approximately 40% among patients with DM, which is higher than the incidence in patients without DM [17]. Patients with DM who also present with AF have increased risks of stroke, cardiovascular disease, mortality, and heart failure [18, 19]. However, the potential bidirectional association between AF and ESRD has not been evaluated, and the role of AF incidence on the progression from DM to ESRD has not been elucidated. Therefore, this study examined the association of AF incidence with the risk of developing ESRD among patients with DM by conducting a retrospective population-based cohort study using claims data from the National Health Insurance Research Database (NHIRD) of Taiwan.

## Materials and methods

### Data source

The National Health Insurance (NHI) program of Taiwan started in 1995 and currently covers nearly 99% of Taiwan's population of approximately 23 million people [20]. Since 1999, the NHI program has made data available to researchers through the NHIRD, which contains registration files and claims data for NHI beneficiaries, including diagnoses, demographics, medication types, prescription dates, dosages, and prescription durations [21]. In the present study, we reviewed claims data for 1 million randomly sampled NHI beneficiaries. The datasets analyzed in the current study are available in the longitudinal health insurance database **(LHID) 2000** repository. The National Health Research Institutes oversees all claims data and generates random identification numbers for insured patients to protect their privacy. Because the NHIRD contains deidentified and anonymously analyzed secondary data, the need for informed consent in this study was waived. This study was approved by the institutional review board of Tungs' Taichung MetroHarbor Hospital in Taichung, Taiwan (106206N). All protocols used in this study were performed in accordance with relevant guidelines and regulations.

From among the 1 million NHI beneficiaries who were randomly sampled from the NHIRD, we first selected those diagnosed with DM according to the International Classification of Diseases, 9th Revision, Clinical Modification (ICD-9-CM; code 250, n = 98,213) and prescribed at least one antidiabetic drug between 2000 and 2013. We divided these patients into those with (n = 8,886) and without AF (n = 89,327) according to ICD-9-CM code 427.31. After excluding those aged <20 years (n = 611) and those diagnosed with ESRD before the index date (n = 1,456), 6,819 patients with DM and AF remained. The AF group was matched on a 1:1 basis from among the 89,327 patients without AF. The date of AF diagnosis among the AF group was defined as the index for both the patients with AF and propensity score–matched non-AF counterparts. All patients were followed until death, any diagnosis of ESRD, or December 31, 2013, whichever came first. A flow chart showing the study patient selection process is presented in Fig 1.

## Definition of research variables

The main outcome of this study was ESRD occurrence, defined as having received dialysis treatment between 2000 and 2013. The main comorbidities that were controlled for in this study were hypertension (ICD-9-CM code 401), heart failure (ICD-9-CM code 428), hyperlipidemia (ICD-9-CM code 272), cardiovascular disease (ICD-9-CM codes 390–459), COPD (ICD-9-CM code 490–496), stroke (ICD-9-CM code 430–438), cancer (ICD-9-CM code 140–208), and chronic kidney disease (CKD; ICD-9-CM code 585). Study patients were considered to have one of these comorbidities if they underwent at least two ambulatory visits and one hospitalization associated with the respective diagnosis.

## Statistical analysis

The selection of patients without AF was performed using a 1:1 propensity score matching method [22], and matching was performed based on the nearest neighbor algorithm with a perfect proportion of 0.995 to 1.0 [23]. Propensity scores for all study patients were calculated using multivariable logistic regression adjusted for age, sex, geographic region, and Charlson Comorbidity Index (CCI) score [24].

The baseline data for patients with and without AF are presented as the frequency and percentage for categorical variables and as the mean and standard deviation for continuous variables. T-tests and $\chi^2$ tests were used to describe differences between groups with and without AF for categorical and continuous variables, respectively. The ESRD incidence rate was defined as the number of events divided by follow-up person-years, which were calculated as the time from the index date to the first received dialysis treatment, death, or December 31, 2013, whichever occurred first. We used Cox proportional hazards regression to determine adjusted hazard ratios (aHRs) and 95% confidence intervals (CIs) for ESRD comparing between groups with and without AF. By conducting the Schoenfeld residual test [25], we confirmed that the proportional hazards assumption was not violated. In multivariable analyses, we adjusted for all covariates shown in Table 2. In order to consider the competing risks of ESRD and death before ESRD, we used the cause-specific hazard model to account for the distribution hazard for competing risks [26]. The Kaplan–Meier model was used to compare ESRD risk between groups with and without AF and between groups stratified according to CKD status (Fig 2). We further performed subgroup analyses to assess differences in AF risk among individual subgroups stratified by sex, age, hypertension, heart failure, dyslipidemia, CAD, COPD, stroke, cancer, CKD, and CCI score. Aside from the stratified variable being examined, all aHRs were adjusted for all other covariates (Fig 3). All $p$-values were two-sided, and any $p$-value <0.05 was considered significant. All analyses were computed using SAS version 9.4 (SAS Institute Inc, Cary, North Carolina).

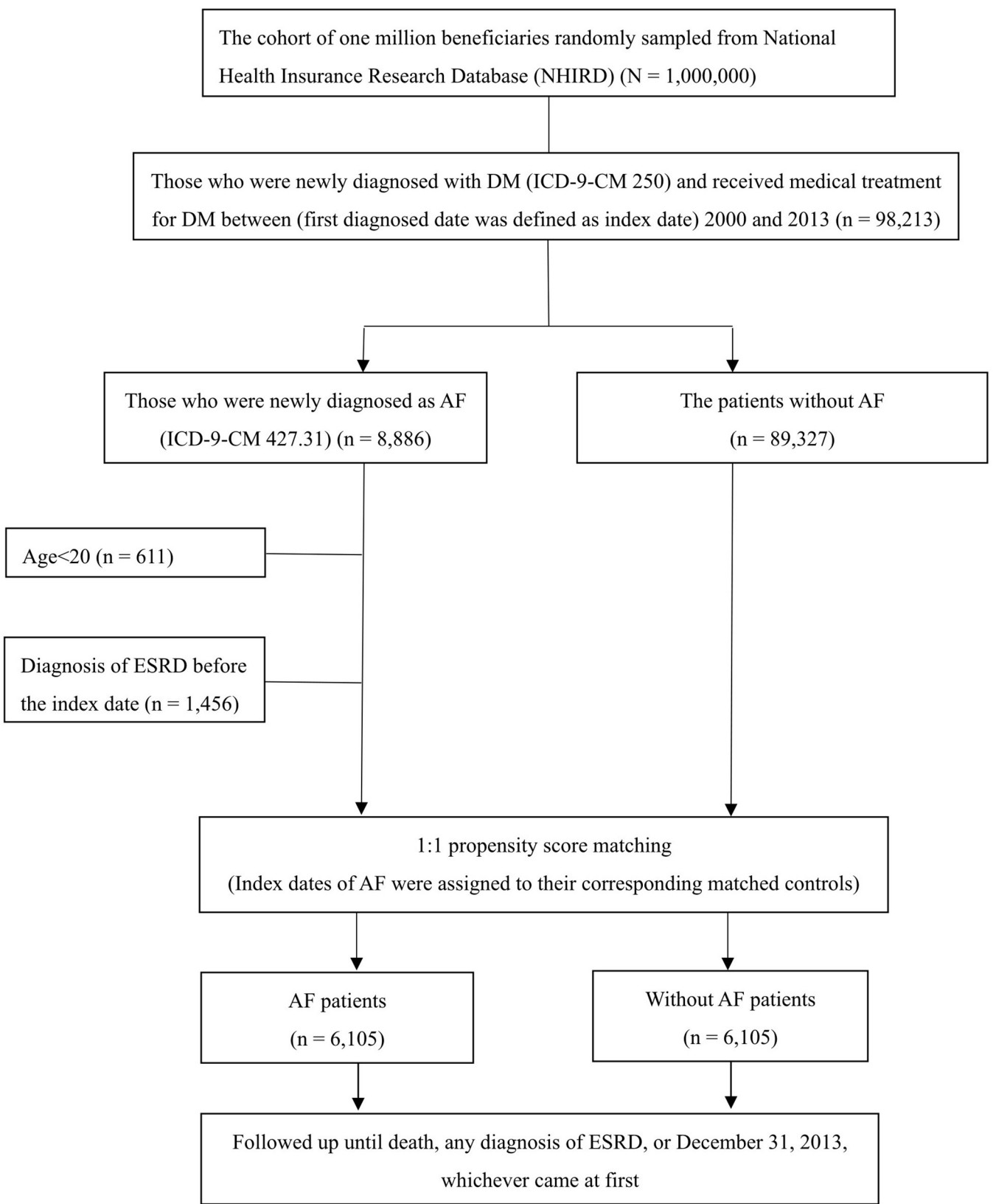

**Fig 1. A flow chart of the patient selection process for this study.**

(a) CKD patients

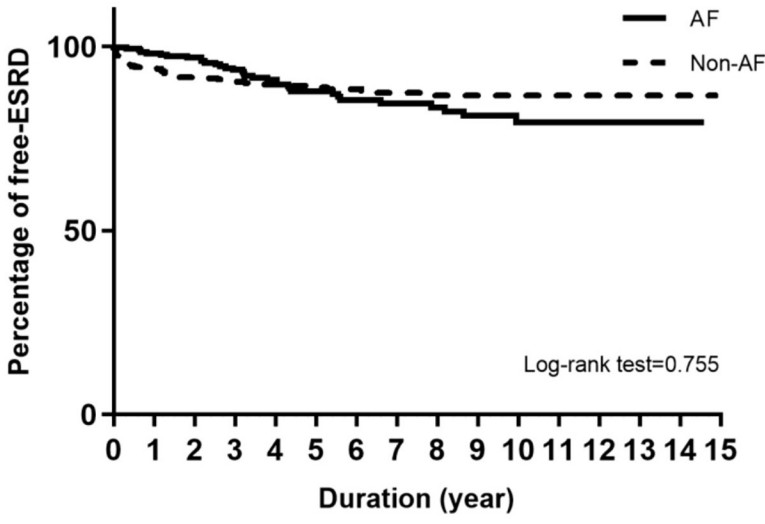

(b) Non-CKD patients

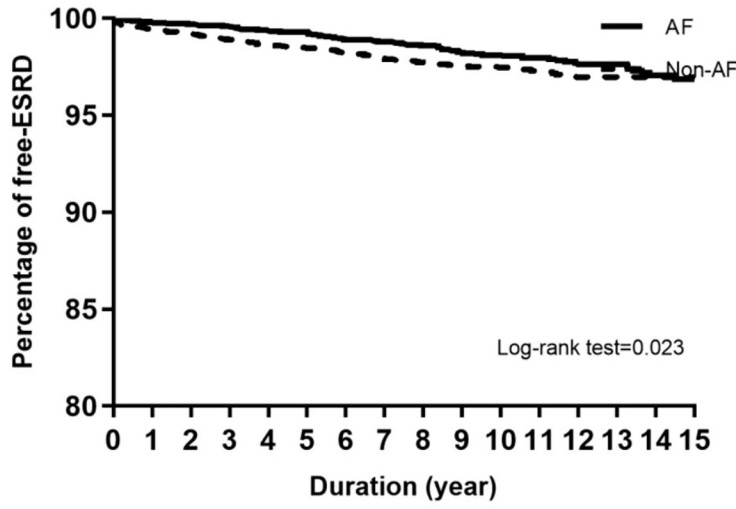

**Fig 2. Comparison of the end-stage renal disease curve between patients with diabetes with and without AF.**

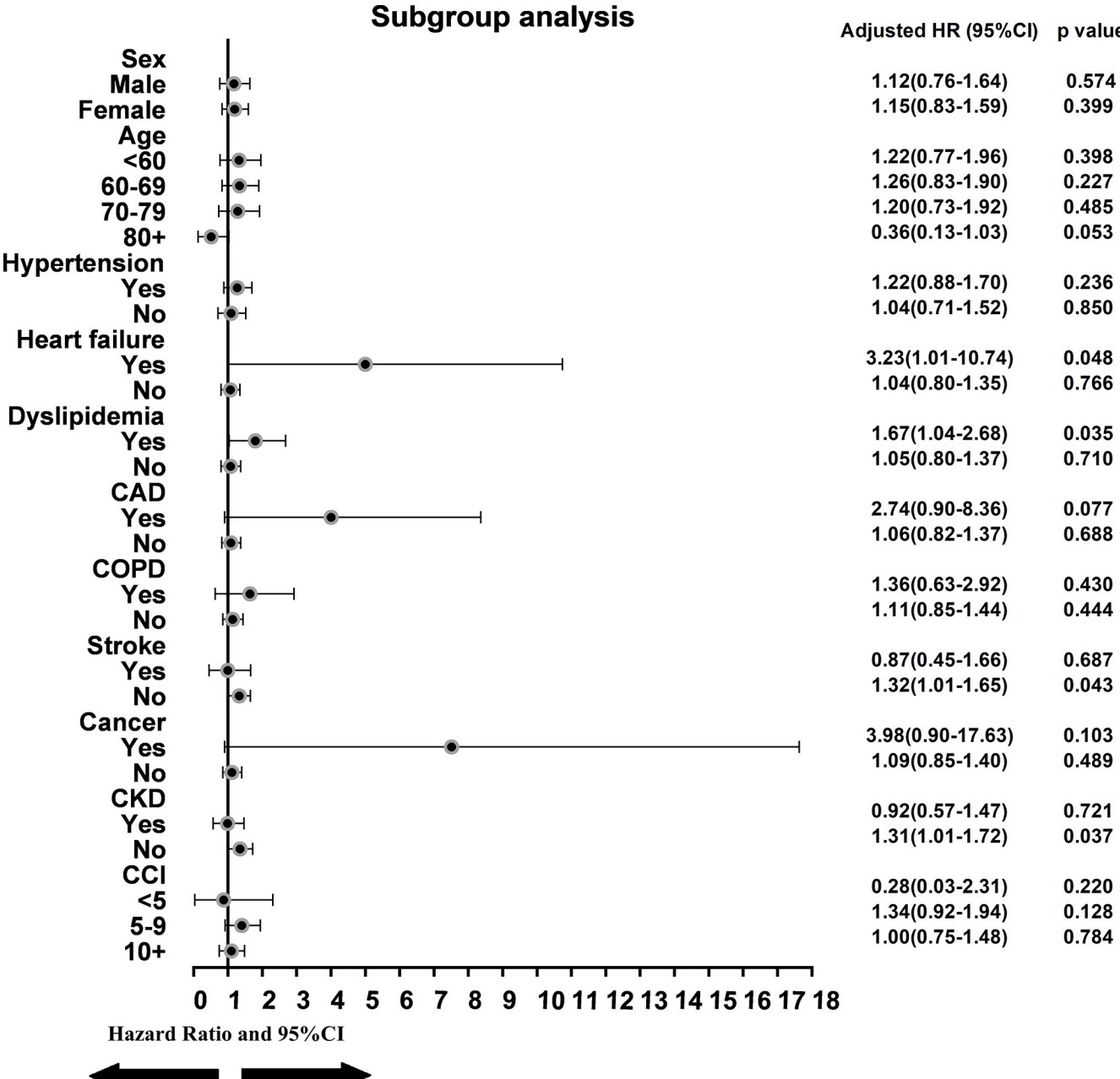

**Fig 3. Risks of developing end-stage renal disease among patients with diabetes with and without atrial fibrillation, stratified according to demographic characteristics and comorbidities.**

## Results

### Basic characteristics of study patients

The baseline characteristics of the patients and their matched controls are shown in Table 1. The average age was 68.7 years, and 51% of patients were men. Compared with DM patients without AF, those with AF were more likely to have hypertension, heart failure, dyslipidemia,

**Table 1. Distribution of demographic characteristics and comorbidities among DM patients with and without AF.**

| | AF | Without AF | |
|---|---|---|---|
| | n (%) | n (%) | *P*-value |
| Sex | n = 6,105 | n = 6,105 | |
| Male | 3,126 (51.2) | 3,138 (51.4) | 0.828 |
| Female | 2,979 (48.8) | 2,967 (48.6) | |
| Age, years | | | |
| <60 | 1,344 (22) | 1,263 (20.7) | 0.169 |
| 60–69 | 1,780 (29.2) | 1,861 (30.5) | |
| 70–79 | 2,087 (34.2) | 2,117 (34.7) | |
| ≥80 | 894 (14.6) | 864 (14.2) | |
| Hypertension | | | |
| Yes | 3,145 (51.5) | 2,075 (34) | <0.001 |
| No | 2,960 (48.5) | 4,030 (66) | |
| Heart failure | | | |
| Yes | 1,197 (19.6) | 279 (4.6) | <0.001 |
| No | 4,908 (80.4) | 5,826 (95.4) | |
| Dyslipidemia | | | |
| Yes | 1,499 (24.6) | 1,229 (20.1) | <0.001 |
| No | 4,606 (75.5) | 4,876 (79.9) | |
| CAD | | | |
| Yes | 573 (9.4) | 281 (4.6) | <0.001 |
| No | 5,532 (90.6) | 5,824 (95.4) | |
| PAOD | | | |
| Yes | 7 (0.1) | 5 (0.1) | 0.564 |
| No | 6,098 (99.9) | 6,100 (99.9) | |
| COPD | | | |
| Yes | 1,171 (19.2) | 637 (10.4) | <0.001 |
| No | 4,934 (80.8) | 5,468 (89.6) | |
| Stroke | | | |
| Yes | 1,278 (20.9) | 761 (12.5) | <0.001 |
| No | 4,827 (79.1) | 5,344 (87.5) | |
| Cancer | | | |
| Yes | 334 (5.5) | 439 (7.2) | <0.001 |
| No | 5,771 (94.5) | 5,666 (92.8) | |
| CKD | | | |
| Yes | 433 (7.1) | 321 (5.3) | <0.001 |
| No | 5,672 (92.9) | 5,784 (94.7) | |
| CCI score, mean ± SD | 9.2 ± 3.5 | 9.2 ± 4.0 | 0.692 |
| <5 | 800 (13.1) | 1,114 (18.3) | <0.001 |
| 5–9 | 3,335 (54.6) | 2,943 (48.2) | |
| ≥10 | 1,970 (32.3) | 2,048 (33.6) | |
| DM duration, mean ± SD | 5.7 ± 4.7 | 5.6 ± 4.8 | 0.056 |
| Duration period, mean ± SD | 7.6 ± 4.6 | 7.5 ± 4.9 | 0.092 |

Abbreviations: AF, atrial fibrillation; CAD, cardiac artery disease; CCI, Charlson Comorbidity Index; CKD, chronic kidney disease; COPD, chronic obstructive pulmonary disorder; DM, diabetes mellitus; PAOD, peripheral arterial occlusive disease.

CAD, COPD, stroke, and CKD; have lower percentages of cancer; and have CCI scores <5. As expected, DM patients with AF had higher prevalences of common comorbidities than those without AF. No significant difference was observed in the duration between the index date and new-onset ESRD between the case (7.6 years) and control groups (7.5 years).

## Hazard ratio estimation for new-onset ESRD

As shown in Table 2, male sex was associated with a lower risk of developing ESRD than female sex (aHR = 0.68, 95% CI = 0.53–0.86, $p$ = 0.015). The risk of developing ESRD decreased (aHR = 0.67, 95% CI = 0.50–0.89, $p$ = 0.007 for 60–69 years; aHR = 0.33, 95% CI = 0.20–0.54, $p < 0.001$ for ≥80 years) with increasing age compared with patients <60 years. AF, hypertension, CKD, and CCI score were strong predictors of ESRD in patients with DM. Compared with DM patients without AF, the those with AF had an increased risk of developing ESRD (aHR = 1.18, 95% CI = 1.01–1.46, $p$ = 0.043). Patients with DM with hypertension had a higher risk of developing ESRD than those with DM without hypertension (aHR = 1.89, 95% CI = 1.46–2.42, $p < 0.001$). Furthermore, the risk of developing ESRD was higher for patients with DM with CKD (aHR = 6.35, 95% CI = 4.80–8.41, $p < 0.001$) than for those with DM without CKD. The risk of developing ESRD increased with increasing CCI scores (aHR = 7.53, 95% CI = 3.07–18.45, $p < 0.001$ for CCI 5–9; aHR = 13.25, 95% CI = 5.39–32.58, $p < 0.001$ for CCI ≥ 10) compared with CCI ≤ 5. Conversely, patients with DM with stroke and cancer were associated with decreased risks of developing ESRD than patients with DM without stroke and cancer. Furthermore, the cause-specific hazard ratio of death before ESRD due to a competing was 1.57 (95% CI, 1.21–3.81).

## Subgroup analyses

The influence of AF status on ESRD among patients with DM was examined by subgroup analyses stratified according to sex (male or female), age (<60, 60–69, 70–79, or ≥80 years), and potential comorbidities (Fig 3). The presence of certain comorbidities significantly increased the risk of developing ESRD among DM patients with AF. Among subgroups of patients with DM with comorbidities, heart failure (aHR = 3.23, 95% CI = 1.02–10.54, $p$ = 0.048) and dyslipidemia (aHR = 1.67, 95% CI = 1.04–2.68, $p$ = 0.035) significantly elevated the risks of developing ESRD compared with those without these comorbidities. However, a distinct difference was observed between DM patients with and without hypertension. Among patients with DM without CKD, those with AF had a higher risk of developing ESRD than those without AF (aHR = 1.31, 95% CI = 1.01–1.72, $p$ = 0.037). When patients with DM were stratified into two groups based on CKD prevalence (Fig 2), the cumulative incidence curve analyzed by the log-rank test showed that among patients with DM without CKD, AF increased the risk of developing ESRD compared with those without AF ($p$ = 0.023), but AF incidence did not appear to have a significant effect on ESRD risk in patients with DM with CKD.

## Discussion

The present population-based study investigated the relationship between atrial fibrillation and ESRD among patients with DM using data from the Taiwan NHIRD and found that AF was associated with a higher risk of subsequent ESRD in patients with DM, and this association was significantly stronger among DM patients without CKD. To the best of the authors' knowledge, this is the first study to investigate the relationship between AF and ESRD among patients with DM in Taiwan.

**Table 2. Association of DM progression to ESRD with AF, sociodemographic characteristics, and comorbidities.**

| | Crude HR (95% CI) | *P*-value | Adjusted HR (95% CI) | *P*-value |
|---|---|---|---|---|
| DM | | | | |
| Without AF | Ref. | | Ref. | |
| AF | 1.36 (1.08–1.83) | 0.011 | 1.18 (1.01–1.46) | 0.043 |
| Sex | | | | |
| Female | Ref. | | Ref. | |
| Male | 0.67 (0.53–0.84) | 0.001 | 0.68 (0.53–0.86) | 0.015 |
| Age, years | | | | |
| <60 | Ref. | | Ref. | |
| 60–69 | 0.81 (0.61–1.08) | 0.809 | 0.67 (0.50–0.89) | 0.013 |
| 70–79 | 0.46 (0.34–0.63) | 0.462 | 0.40 (0.29–0.55) | <0.001 |
| ≥80 | 0.32 (0.20–0.51) | 0.320 | 0.33 (0.20–0.54) | <0.001 |
| DM duration | 1.02 (0.98–1.05) | 0.267 | 0.99 (0.94–1.03) | 0.117 |
| Hypertension | | | | |
| No | Ref. | | Ref. | |
| Yes | 2.32 (1.82–2.95) | <0.001 | 1.89 (1.46–2.42) | <0.001 |
| Heart failure | | | | |
| No | Ref. | | Ref. | |
| Yes | 1.52 (1.11–2.09) | 0.009 | 1.15 (0.83–1.60) | 0.415 |
| Dyslipidemia | | | | |
| No | Ref. | | Ref. | |
| Yes | 2.07 (1.61–2.67) | <0.001 | 1.17 (0.89–1.52) | 0.262 |
| CAD | | | | |
| No | Ref. | | Ref. | |
| Yes | 1.28 (0.84–1.97) | 0.253 | 0.91 (0.59–1.41) | 0.677 |
| COPD | | | | |
| No | Ref. | | Ref. | |
| Yes | 0.76 (0.53–1.10) | 0.146 | 0.67 (0.44–1.02) | 0.068 |
| Stroke | | | | |
| No | Ref. | | Ref. | |
| Yes | 0.89 (0.64–1.25) | 0.892 | 0.62 (0.44–0.90) | 0.029 |
| Cancer | | | | |
| No | Ref. | | Ref. | |
| Yes | 0.64 (0.34–1.20) | 0.637 | 0.46 (0.25–0.88) | 0.046 |
| CKD | | | | |
| No | Ref. | | Ref. | |
| Yes | 8.51 (6.54–11.06) | <0.001 | 6.35 (4.80–8.41) | <0.001 |
| CCI score | | | | |
| <5 | Ref. | | Ref. | |
| 5–9 | 7.65 (3.13–18.69) | <0.001 | 7.53 (3.07–18.45) | <0.001 |
| ≥10 | 12.74 (5.22–31.07) | <0.001 | 13.25 (5.39–32.58) | <0.001 |

Abbreviations: AF, atrial fibrillation; CAD, cardiac artery disease; CCI, Charlson Comorbidity Index; CI, confidence interval; CKD, chronic kidney disease; COPD, chronic obstructive pulmonary disorder; DM, diabetes mellitus; ESRD, end-stage renal disease; HR, hazard ratio; Ref., reference.

The primary new finding of this study is that AF is an important predictor of ESRD in patients with DM. Previous studies have reported that DM is an independent risk factor for AF [27–29], and metabolic syndrome is hypothesized to increase epicardial adipose tissue

thickness, leading to the release of pro-inflammatory substances that contribute the endothelial dysfunction and fibrosis and influence structural and electrical atrial remodeling [30, 31]. However, no previously published studies have evaluated the impacts of incident AF on the progression from DM to adverse renal events. Our study of a large community-based cohort of patients with DM found that incident AF was associated with a higher rate of ESRD, suggesting that incident AF contributes to progression from DM to ESRD.

Multiple biological mechanisms may underlie the accelerated progression to ESRD from DM in patients with AF. First, AF triggers systemic inflammation [32]. A cohort study showed that ESRD risk in patients with DM was strongly associated with elevated concentrations of circulating pro-inflammatory cytokines, which might be responsible for kidney dysfunction [33]. Second, increasing evidence supports an association between inflammation and AF [34–36]. Additionally, various inflammatory biomarkers have been associated with AF, suggesting an important role for inflammation in AF [34], resulting in a prothrombotic state that induces renal micro-infarcts, decreased renal function, and rapid progression from DM to ESRD. Third, AF reduces left ventricular systolic and diastolic function, which may accelerate ESRD progression and impair kidney function through altered hemodynamics, reduced renal perfusion, and renin-angiotensin-aldosterone system activation [37, 38]. Fourth, some medications used to treat AF are nephrotoxic and can result in renal impairment.

We found that the association of incident AF with ESRD risk was stronger among DM patients without CKD. This finding is particularly interesting because CKD is generally accepted to increase the risk of developing AF [39–41]. The reasons for this finding are unclear; however, AF may result in stronger impacts among patients with DM without CKD because these patients are less likely to have other concomitant comorbidities that contribute to ESRD risk. Further studies are warranted to clarify this finding.

Patients with DM with AF were significantly more likely to have comorbid conditions compared with the patients without AF, including hypertension, heart failure, dyslipidemia, CAD, COPD, stroke, and CKD (Table 1). Remarkably, elevated blood pressure has been known to increase the risks of ESRD occurrence [42, 43]. As shown in univariate analyses (Table 2), hypertension plays a significant role in ESRD occurrence. However, in subgroup analyses according to the presence of hypertension, incident AF was not significantly associated with ESRD among DM patients with hypertension. The reasons for this are complex. For example, the role of hypertension as a risk factor may be masked by successful strategies designed to control blood pressure. However, data regarding the use of blood pressure medications were not available in the Taiwan NHIRD.

The present study has some limitations. First, individual behaviors associated with ESRD development were not able to be evaluated, and not all clinical data were fully available in the Taiwan NHIRD. Thus, we were not able to adjust for other potential confounding factors, such as relevant medications, blood pressure, residual renal function, and hemoglobin level. Second, all diagnoses were based on ICD-9-CM diagnosis codes, which do not differentiate among AF types and pose a risk for coding bias. Lastly, the results generated from our study may not be applicable to other countries as the study was conducted in only one country.

## Conclusions

In conclusion, incident AF is associated with a relative increase in the risk of ESRD development among patients with DM. Further research remains necessary to delineate contributing factors that lead to AF development in the setting of DM and elucidate potential modifiable pathways through which AF contributes to the progression to ESRD.

## Supporting information

**S1 Checklist. STROBE statement.**
(DOC)

## Author Contributions

**Conceptualization:** Yu-Kang Chang, Hueng-Chuen Fan, Wan-Ni Tsai, Paik-Seong Lim.

**Data curation:** Yu-Kang Chang, Wan-Ni Tsai, Paik-Seong Lim.

**Formal analysis:** Yu-Kang Chang.

**Investigation:** Yu-Kang Chang, Wan-Ni Tsai, Paik-Seong Lim.

**Methodology:** Yu-Kang Chang, Yuan-Hung Wang, Paik-Seong Lim.

**Project administration:** Wan-Ni Tsai.

**Supervision:** Paik-Seong Lim.

**Validation:** Hueng-Chuen Fan, Paik-Seong Lim.

**Visualization:** Hueng-Chuen Fan.

**Writing – original draft:** Yu-Kang Chang, Chi-Chien Lin, Yuan-Hung Wang, Wan-Ni Tsai, Paik-Seong Lim.

**Writing – review & editing:** Chi-Chien Lin, Yuan-Hung Wang, Wan-Ni Tsai, Paik-Seong Lim.

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
