## [Decision Letter · Decision Letter 0]

20 Jan 2022

PONE-D-21-17026

Association between Atrial Fibrillation and Risk of End-Stage Renal Disease among Adults with Diabetes Mellitus

PLOS ONE

Dear Dr. Lim,

Thank you for submitting your manuscript to PLOS ONE. After careful consideration, we feel that it has merit but does not fully meet PLOS ONE’s publication criteria as it currently stands. Therefore, we invite you to submit a revised version of the manuscript that addresses the points raised during the review process.

We look forward to receiving your revised manuscript.

Kind regards,

Abduzhappar Gaipov

Academic Editor

PLOS ONE

Additional Editor Comments (if provided):

The methodological issues should be improved

Journal Requirements:

2. Thank you for stating the following financial disclosure: "No"

"This work was funded by grants from Tungs’ Taichung Metroharbor Hospital (TTMHH-106R)."

"This work was funded by grants from Tungs’ Taichung Metroharbor Hospital (TTMHH-106R)."

Reviewers' comments:

Reviewer's Responses to Questions

**Comments to the Author**

1. Is the manuscript technically sound, and do the data support the conclusions?

Reviewer #1: Yes

Reviewer #2: No

2. Has the statistical analysis been performed appropriately and rigorously? 

Reviewer #1: Yes

Reviewer #2: No

3. Have the authors made all data underlying the findings in their manuscript fully available?

Reviewer #1: No

Reviewer #2: No

4. Is the manuscript presented in an intelligible fashion and written in standard English?

Reviewer #1: Yes

Reviewer #2: Yes

5. Review Comments to the Author

Reviewer #1: 1) As authors evaluate the effect of AF in progression of ESRD among DM patients, it would be useful to have a discussion

about the prevalence of AF among DM patients. Authors had discussed the joint effect of AF and ESRD on mortality and morbidity,

but the focus on DM patients is missing in introduction. Are the discussed excessive risk is similar for DM patients or different?

2) Authors declare taking age, gender, region and CCI score for calculation of PS score, but justification on variable selection

for Propensity score analysis is missing. Was it based on statistical methods or clinical importance? Did author have chance to adjust

on more number of variables at this stage? Adjusting for duration of DM or severity at index date might have introduced better

similarity among groups.

3) As authors conducted the retrospective cohort study, thorough explanation of dates is important. At what point the age was taken? Index date, the onset of DM or end of follow-up?

The explanation of defining index date is unclear in "Methods" part. Was it taken after PS matching and individual date of AF incidence

was taken for specific controls? If it is true, this have some drawbacks. Despite the age being controlled, the onset of DM might be different, unless you control it,

but there was no indication on adjustment on DM onset date or its duration. Thus, taking the date of AF onset as an index date for controls might introduce bias

as those with AF might tend to have longer DM duration compared to those without and those with AF will have higher risk of development of other complications at

baseline. Suggestion is again to control for DM duration and if possible the age of DM onset. Also, changing the index date to the date of DM onset might reduce bias.

Additionally, the way of counting the mentioned comorbidities in the study is vague. Was it counted throughout the 2000-2013 period or only after the index date?

If the information on comorbidities' incidence before index date is available, these information might have been used to PS score calculation to make comparison

group homogeneous regarding baseline comorbidities.

4) As there patients might die before developing ESRD, it might worth to perform competing risk analysis.

5) It would be better to include the initial distribution of demographic characteristics and comorbidities before PS matching to be able to evaluate on the goodness of

balancing and how distribution of comorbidities changes.

Reviewer #2: Thank you for sharing your manuscript with me for review. In general the study has an interesting research question and has sufficiently large study sample to explore the research question. However, there are several major limitations in statistical analysis and description of the methods that question methodological robustness of the study. Please my comments below:

1. I appreciate that the authors are attempting to examine the association between AF and ESRD among DM patients where the exposure variable is AF and ESRD is the outcome. However, the following statement in the introduction section is a bit confusing: “…Furthermore, the impact of AF incidence on the progression of DM to ESRD has not been elucidated.”

2. Not familiar with the term “insurance cohort study”. I would recommend to omitting “insurance” word from the study design description as it could cause confusion.

3. In this sentence, “All protocols used in this study were performed in accordance with relevant guidelines and regulations”, could the authors clarify what does it mean relevant guidelines and regulations?

4. In this sentence, “… we first selected those who had been diagnosed with DM according to the International Classification of Diseases, 9th Revision, Clinical Modification (ICD-9-CM; code 250, n = 98,213) and at least one antidiabetic drug”, what did author mean by “at least one antidiabetic drug”?

5. I am not familiar with the NHIRD, but I am guessing it could contain diagnoses about comorbidities. Have the authors attempted to look for AF and DM in all data including among comorbidities or variables that could point to these diseases? For example, elevated blood glucose level could be indication for DM or ECG records for AF?

6. It would be nice to see the definitions of the ICD codes that were used in the manuscript. At least in the Supplementary section.

7. This sentence is not clear “After excluding those aged <20 years (n = 611) and diagnosed with TMD before the index date (n = 1,456) among the AF group, patients were then matched on a 1:1 basis with 89,327 patients without AF, with the initial date of AF diagnosis for a given AF subject being defined as the index date for the without AF group with which he or she was matched.”

8. What does “TMD” stand for?

9. Clearly define what index date means in this context.

10. The description of the propensity score matching approach should be placed in the statistical analysis subsection.

11. Did the authors adjusted for “age, gender, geographic region, and Charlson Comorbidity Index (CCI)” or did they use these variables to calculate propensity scores? Why were these variables selected for the propensity score calculation?

12. Why did the authors choose to include the covariates in Table 2 when building the multivariable model? Were other covariates available from the NHIRD database that could potentially confound the relationship of AF with ESRD?

13. Since death is a competing risk for ESRD occurrence, the authors could perform additionally competing risk regression analysis. I would strongly encourage to perform such analysis.

14. It is not clear why the researchers included some of the variables in the multivariable model in Table 2 that were used for propensity score calculations such as age, sex and CCI? Have the authors determined whether patients with AF and without AF have similar distributions in terms of the variables that were used to calculate the propensity scores?

15. In the subgroup analyses, it is not clear how adjusted HRs were calculated. Did the authors include all interactions together in one model or did they calculate estimates by including one interaction at a time?

16. Why were the confidence intervals for heart failure, CAD and cancer too wide? Have the authors tried to minimize numerical issues when calculating HRs with such wide confidence intervals?

17. It would be also interesting to see if unmatched groups had similar results when performing the multivariable regression analyses? The authors could provide as additional table in the Supplementary section.

18. In Table 1, CCI score has two p-values. Is this a typo or the authors performed two tests?

19. The first two sentences in the Results section are redundant and can be omitted.

20. The following statement “that AF in an important predictor of ESRD in patients with DM” is too bold and not supported given limitations of the study as the authors pointed out some of the important covariates are missing from the multivariable analysis. The authors should avoid such questionable claims and frame the discussion around the association, not causality.

21. What did the authors mean by “pose a risk for coding bias”? Measurement errors?

6. PLOS authors have the option to publish the peer review history of their article (what does this mean?). If published, this will include your full peer review and any attached files.

Reviewer #1: No

Reviewer #2: No

---

## [Author Response · Author response to Decision Letter 0]

21 Apr 2022

Response to Reviewers’ comments

Ref.: PONE-D-21-17026

Reviewers’ comments:

Reviewer #1:

Reviewer #1: 1) As authors evaluate the effect of AF in progression of ESRD among DM patients, it would be useful to have a discussion about the prevalence of AF among DM patients. Authors had discussed the joint effect of AF and ESRD on mortality and morbidity, but the focus on DM patients is missing in introduction. Are the discussed excessive risk is similar for DM pati ents or different?

Reply: Thank you for your comment. Please see Main document_Clean file. We have added the following sentences and associated references to the introduction (P3L25–26 and P4L1–3): “AF is associated with poor outcomes among patients with ESRD [16], and the incidence of AF was approximately 40% among patients with DM, which is higher than the incidence in patients without DM [17]. Patients with DM who also present with AF have increased risks of stroke, cardiovascular disease, mortality, and heart failure [18, 19].”

2) Authors declare taking age, gender, region and CCI score for calculation of PS score, but justification on variable selection for Propensity score analysis is missing. Was it based on statistical methods or clinical importance? Did author have chance to adjust on more number of variables at this stage? Adjusting for duration of DM or severity at index date might have introduced better similarity among groups.

Reply: Thank you for your comment. We selected age, sex, geographic region, and CCI score as the variables included in the propensity score because the inclusion of additional risk factors would result in the loss of many samples and representation. We opted to use the CCI score before the index date to control for the presence of comorbidities during matching.

3) As authors conducted the retrospective cohort study, thorough explanation of dates is important. At what point the age was taken? Index date, the onset of DM or end of follow-up? The explanation of defining index date is unclear in “Methods” part. Was it taken after PS matching and individual date of AF incidence was taken for specific controls? If it is true, this have some drawbacks. Despite the age being controlled, the onset of DM might be different, unless you control it, but there was no indication on adjustment on DM onset date or its duration. Thus, taking the date of AF onset as an index date for controls might introduce bias as those with AF might tend to have longer DM duration compared to those without and those with AF will have higher risk of development of other complications at baseline. Suggestion is again to control for DM duration and if possible the age of DM onset. Also, changing the index date to the date of DM onset might reduce bias. Additionally, the way of counting the mentioned comorbidities in the study is vague. Was it counted throughout the 2000-2013 period or only after the index date? If the information on comorbidities’ incidence before index date is available, these information might have been used to PS score calculation to make comparison group homogeneous regarding baseline comorbidities.

Reply: Thank you for your comment. The index date was the date of AF incidence among patients with DM with AF, and this date was used as the index date for all matched patients with DM without AF. The index date was defined in the Methods as follows: “The date of AF diagnosis among the AF group was defined as the index for both the patients with AF and propensity score–matched non-AF counterparts.” (Please see P5L9–10). The study was focused on patients with and without AF among patients with DM, and all patients were followed from the index date until death, any ESRD diagnosis, or December 31, 2013, whichever came first. Although we selected the date of AF diagnosis as the index day, we included DM duration and the age of DM diagnoses as adjusted variables (Please see revised Tables 1 and 2 and Figure 3, P7L11–14 and P7L17–24 and P8L7–8 and P8L12–13 in the Main document_Clean file).

4) As there patients might die before developing ESRD, it might worth to perform competing risk analysis.

Reply: Thank you for your comment. The adjusted hazard ratio of death before ESRD due to a competing was 1.57 (95% CI, 1.21–3.81), which was similar to the original hazard ratio.

5) It would be better to include the initial distribution of demographic characteristics and comorbidities before PS matching to be able to evaluate on the goodness of balancing and how distribution of comorbidities changes.

Reply: Thank you for your comment. We performed propensity score matching based on sex, age, and CCI to control the distribution of demographic characteristics among patients with DM with and without AF.

Reviewer #2: Thank you for sharing your manuscript with me for review. In general the study has an interesting research question and has sufficiently large study sample to explore the research question. However, there are several major limitations in statistical analysis and description of the methods that question methodological robustness of the study. Please my comments below:

1. I appreciate that the authors are attempting to examine the association between AF and ESRD among DM patients where the exposure variable is AF and ESRD is the outcome. However, the following statement in the introduction section is a bit confusing: “…Furthermore, the impact of AF incidence on the progression of DM to ESRD has not been elucidated.”

Reply: Thank you for your comment. We have revised the sentence as follows. “However, the potential bidirectional association between AF and ESRD has not been evaluated, and the role of AF incidence on the progression from DM to ESRD has not been elucidated.” (Please see P4L3–5 in the Main document_Clean file)

2. Not familiar with the term “insurance cohort study”. I would recommend to omitting “insurance” word from the study design description as it could cause confusion.

Reply: Thank you for your comment. We had have removed the word “insurance” and appreciate your recommendation. (Please see P4L7 in the Main document_Clean file)

3. In this sentence, “All protocols used in this study were performed in accordance with relevant guidelines and regulations”, could the authors clarify what does it mean relevant guidelines and regulations?

Reply: This sentence means this study was performed in accordance with the institutional review board guidelines and regulations.

4. In this sentence, “… we first selected those who had been diagnosed with DM according to the International Classification of Diseases, 9th Revision, Clinical Modification (ICD-9-CM; code 250, n = 98,213) and at least one antidiabetic drug”, what did author mean by “at least one antidiabetic drug”?

Reply: Because all individuals were categorized as having DM based only on data available in the NHIRD, we defined the criteria for DM as having both a DM diagnosis code and the use of at least one antidiabetic drug to ensure our cohort was appropriately diagnosed.

5. I am not familiar with the NHIRD, but I am guessing it could contain diagnoses about comorbidities. Have the authors attempted to look for AF and DM in all data including among comorbidities or variables that could point to these diseases? For example, elevated blood glucose level could be indication for DM or ECG records for AF?

Reply: The NHIRD contains diagnosis codes for AF, DM, and comorbidities, and we attempted to evaluate the relationship between AF and ESRD among DM patients after adjusting for comorbidities.

6. It would be nice to see the definitions of the ICD codes that were used in the manuscript. At least in the Supplementary section.

Reply: Thank you for your comment. We have provided definitions of the ICD codes in the Methods section. (Please see P5L3, P5L6 and P5L17–21 in the Main document_Clean file)

7. This sentence is not clear “After excluding those aged <20 years (n = 611) and diagnosed with TMD before the index date (n = 1,456) among the AF group, patients were then matched on a 1:1 basis with 89,327 patients without AF, with the initial date of AF diagnosis for a given AF subject being defined as the index date for the without AF group with which he or she was matched.”

Reply: Thank you for your comment. We have revised this sentence as follows: “After excluding those aged <20 years (n = 611) and those diagnosed with ESRD before the index date (n = 1,456), 6,819 patients with DM and AF remained. The AF group was matched on a 1:1 basis from among the 89,327 patients without AF. The date of AF diagnosis among the AF group was defined as the index for both the patients with AF and propensity score–matched non-AF counterparts.” (Please see P5L6–10 in the Main document_Clean file)

8. What does “TMD” stand for?

Reply: Thank you for your comment. The “TMD” was an error and has been revised to “ESRD.” (Please see P5L7 in the Main document_Clean file, and Figure 1)

9. Clearly define what index date means in this context.

Reply: The index date was the date of AF incidence among patients with DM with AF, and this date was used as the index date for all matched patients with DM without AF. (Please see P5L9–10 in the Main document_Clean file)

10. The description of the propensity score matching approach should be placed in the statistical analysis subsection.

Reply: Thank you for your recommendation. The description of the propensity score matching approach has been placed in the first section of the statistical analysis section. (Please see P5L25–26 and P6L1-3 in the Main document_Clean file)

11. Did the authors adjusted for “age, gender, geographic region, and Charlson Comorbidity Index (CCI)” or did they use these variables to calculate propensity scores? Why were these variables selected for the propensity score calculation?

Reply: Thank you for your comment. We used the propensity score matching based on the sex, age, and Charlson Comorbidity Index (CCI) score because the inclusion of too many risk factors would result in the loss of numerous samples and representation. We opted to use the CCI score before the index date to control for comorbidities during the matching process.

12. Why did the authors choose to include the covariates in Table 2 when building the multivariable model? Were other covariates available from the NHIRD database that could potentially confound the relationship of AF with ESRD?

Reply: Thank you for your comment. We selected covariates that could be potential confounding factors in the progression to ESRD. Unfortunately, some potentially confounding factors that might contribute to ESRD are not available in the NHIRD, including individual behaviors, some clinical data, relevant medications, blood pressure, residual renal function, and hemoglobin levels. We have added this as a limitation of our study in the last section of the Discussion (Please see P10L16–20 in the Main document_Clean file).

13. Since death is a competing risk for ESRD occurrence, the authors could perform additionally competing risk regression analysis. I would strongly encourage to perform such analysis.

Reply: Thank you for your comment. The adjusted hazard ratio of death before ESRD due to a competing was 1.57 (95% CI, 1.21–3.81), which was similar to the original hazard ratio.

14. It is not clear why the researchers included some of the variables in the multivariable model in Table 2 that were used for propensity score calculations such as age, sex and CCI? Have the authors determined whether patients with AF and without AF have similar distributions in terms of the variables that were used to calculate the propensity scores?

Reply: We performed propensity score matching to identify patients with AF and without AF with similar demographics (sex and age) and comorbidities (CCI) to provide a better comparison between these two groups.

15. In the subgroup analyses, it is not clear how adjusted HRs were calculated. Did the authors include all interactions together in one model or did they calculate estimates by including one interaction at a time?

Reply: Thank you for your comment. We compared individual subgroups stratified by a single variable at a time, including sex, age, hypertension, heart failure, dyslipidemia, CAD, COPD, stroke, cancer, CKD, and CCI score. Except for the examined variable, all adjusted HRs were adjusted for all covariates. We have revised this explanation in the manuscript. (Please see P6L17-21 in the Main document_Clean file)

16. Why were the confidence intervals for heart failure, CAD and cancer too wide? Have the authors tried to minimize numerical issues when calculating HRs with such wide confidence intervals?

Reply: Thank you for your comment. Because of propensity score matching, the numbers of patients with heart failure, CAD, or cancer were fewer than patients with other comorbidities, and these subgroups showed a reduction in ESRD incidence but with wide confidence intervals.

17. It would be also interesting to see if unmatched groups had similar results when performing the multivariable regression analyses? The authors could provide as additional table in the Supplementary section.

Reply: Thank you for your comment. In our study design, the index date for non-AF patients was taken at the time of matched AF-patient onset. The observation period began from the index date to the final follow-up date, which is why non-AF (patients) were not subjected to Cox proportional regression analysis and lack matching index dates. As a result, we are unable to provide the results of unmatched groups.

18. In Table 1, CCI score has two p-values. Is this a typo or the authors performed two tests?

Reply: Thank you for your comment. The first p-value is the comparison of continuous variables by t-test. The second p-value is the comparison of categorical variables by the chi-square test.

19. The first two sentences in the Results section are redundant and can be omitted.

Reply: Thank you for your comment. We have deleted these first two sentences (Please see the Main document_Clean file).

20. The following statement “that AF in an important predictor of ESRD in patients with DM” is too bold and not supported given limitations of the study as the authors pointed out some of the important covariates are missing from the multivariable analysis. The authors should avoid such questionable claims and frame the discussion around the association, not causality.

Reply: Thank you for your comment. We had revied the last sentence of the Discussion to describe the finding of an association between incident AF and ESRD (Please see P11L1-2 in the Main document_Clean file).

21. What did the authors mean by “pose a risk for coding bias”? Measurement errors?

Reply: Thank you for your comment. Because NHIRD data represents health insurance declaration data, a very small chance of coding bias exists.

---

## [Decision Letter · Decision Letter 1]

6 Jun 2022

PONE-D-21-17026R1Association Between Atrial Fibrillation and Risk of End-Stage Renal Disease Among Adults With Diabetes MellitusPLOS ONE

Dear Dr. Lim,

Thank you for submitting your manuscript to PLOS ONE. After careful consideration, we feel that it has merit but does not fully meet PLOS ONE’s publication criteria as it currently stands. Therefore, we invite you to submit a revised version of the manuscript that addresses the points raised during the review process.

We look forward to receiving your revised manuscript.

Kind regards,

Abduzhappar Gaipov

Academic Editor

PLOS ONE

Reviewers' comments:

Reviewer's Responses to Questions

**Comments to the Author**

1. If the authors have adequately addressed your comments raised in a previous round of review and you feel that this manuscript is now acceptable for publication, you may indicate that here to bypass the “Comments to the Author” section, enter your conflict of interest statement in the “Confidential to Editor” section, and submit your "Accept" recommendation.

Reviewer #1: All comments have been addressed

Reviewer #2: (No Response)

2. Is the manuscript technically sound, and do the data support the conclusions?

Reviewer #1: Yes

Reviewer #2: Partly

3. Has the statistical analysis been performed appropriately and rigorously? 

Reviewer #1: Yes

Reviewer #2: No

4. Have the authors made all data underlying the findings in their manuscript fully available?

Reviewer #1: No

Reviewer #2: No

5. Is the manuscript presented in an intelligible fashion and written in standard English?

Reviewer #1: Yes

Reviewer #2: Yes

6. Review Comments to the Author

Reviewer #1: Thank you for letting me review given article. I have no further comments. Comments were addressed as fully as possible.

Reviewer #2: I appreciate that the authors' responses to my comments. However, I still have a few concerns that might be helpful in strengthening the manuscript:

The authors indicated that some risk factors have missing values which result in smaller sample size and a loss of statistical power. Given presence of missing values of important covariates for analysis, it would be interesting to conduct multiple imputations (MI). MI will allow a better statistical power and adjustment for more appropriate set of covariates to draw conclusions.

I see that the authors performed competing risk analysis. I would suggest including the description of methods used to perform the competing risk analysis and provide these results in the manuscript.

While the authors indicated that groups were matched using PS, Table 1 shows that there are differences in some of the important covariates. I do understand that the authors included these covariates later in the multivariable regression. However, why would the authors perform PS matching (while missing some of important covariates) and then including them in the final analysis. Would it be easier to just conduct multivariable analysis without matching? And then conduct analysis on the matched groups? I am just struggling to understand with concepts why the authors done the analysis this way.

7. PLOS authors have the option to publish the peer review history of their article (what does this mean?). If published, this will include your full peer review and any attached files.

Reviewer #1: No

Reviewer #2: No

---

## [Author Response · Author response to Decision Letter 1]

4 Jul 2022

Reviewer #1: Thank you for letting me review given article. I have no further comments. Comments were addressed as fully as possible.

Reply: Thank you.

Reviewer #2: I appreciate that the authors' responses to my comments. However, I still have a few concerns that might be helpful in strengthening the manuscript:

The authors indicated that some risk factors have missing values which result in smaller sample size and a loss of statistical power. Given presence of missing values of important covariates for analysis, it would be interesting to conduct multiple imputations (MI). MI will allow a better statistical power and adjustment for more appropriate set of covariates to draw conclusions.

Reply: Thank you for your comment. MI is a tool to deal the missing research data essentially that happens when people fail to respond to a survey. The NHIRD is a registered claims dataset that has fewer missing data problem. In the previous study was showed that a rule of thumb blow of 5% missingness was not necessary to multiple imputation.1 So that MI might not be suitable to use in the study that is not somewhat true to the original in order to add the statistic power. On the other hand, we chose the PS matching in the statistical analysis of observational data to construct an artificial control group by matching each treated unit with a non-treated unit of similar characteristics that attempts to reduce bias arises. 

Reference: 

1.Jakobsen JC, Gluud C, Wetterslev J, Winkel P. When and how should multiple imputation be used for handling missing data in randomised clinical trials - a practical guide with flowcharts. BMC Med Res Methodol. 2017;17(1):162. Epub 2017/12/07. doi: 10.1186/s12874-017-0442-1. PubMed PMID: 29207961; PubMed Central PMCID: PMCPMC5717805.

I see that the authors performed competing risk analysis. I would suggest including the description of methods used to perform the competing risk analysis and provide these results in the manuscript.

Reply: Thank you for your comment. Please see Main document_Clean file. We have add the description of competing risk analysis in the Methods. (Please see P6L15-17)

While the authors indicated that groups were matched using PS, Table 1 shows that there are differences in some of the important covariates. I do understand that the authors included these covariates later in the multivariable regression. However, why would the authors perform PS matching (while missing some of important covariates) and then including them in the final analysis. Would it be easier to just conduct multivariable analysis without matching? And then conduct analysis on the matched groups? I am just struggling to understand with concepts why the authors done the analysis this way.

Reply: Thank you for your kindly comment. In the statistical analysis of observational data, PS matching is a quasi-experimental method such as the investigator uses statistical techniques to construct an artificial control group by matching each treated unit with a non-treated unit of similar characteristics that attempts to reduce bias arises. Because a difference in the treatment outcome between treated and untreated groups may be caused by a factor that predicts treatment rather than the treatment itself. We had used this statistical analysis published some journal articles based on NHIRD1-8. In our earliest study8, we used PS matching among patients with peritoneal dialysis and hemodialysis, because of patients treated with PD were different from patients treated with HD in terms of health status. The PD and HD cohorts were homogenous in sociodemographic characteristics and comorbid conditions at the baseline after using the propensity score-matched design, which reduces the selection bias of samples and potential confounding effect. The general rule of practice in the previous study was showed that the covariates can be added into a regression adjustment that could be dramatically remove residual confounding bias if remaining imbalance after PS matching.9

References:

1.Chen YY, Fan HC, Tung MC, Chang YK. The association between Parkinson's disease and temporomandibular disorder. PLoS One. 2019; 14(6):e0217763.

2.Lu CW, Chang YK, Lee YH, Kuo CS, Chang HH, Huang CT, Hsu CC, Huang KC. Increased Risk for Major Depressive Disorder in Severely Obese Patients after Bariatric Surgery - a 12-year Nationwide Cohort Study. Ann Med. 2018; 13:1-30. doi: 10.1080/07853890.2018.1511917.

3.Chang HH, Chang YK, Lu CW, Huang CT, Chien CT, Hung KY, Huang KC, Hsu CC. Statins Improve Long Term Patency of Arteriovenous Fistula for Hemodialysis. Sci Rep. 2016; 6:22197. doi: 10.1038/srep22197.

4.Lu CW, Chang YK, Chang HH, Kuo CS, Huang CT, Hsu CC, Huang KC. Fracture Risk After Bariatric Surgery: A 12-Year Nationwide Cohort Study. Medicine (Baltimore). 2015; 94(48): e2087.

5.Hung SC, Chang YK, Liu JS, Hsu CC, Tarng DC. Mortality and metformin use in patients with advanced chronic kidney disease. Lancet Diabetes Endocrinol. 2015 Aug;3(8):605-14. doi: 10.1016/S2213-8587(15)00123-0. 

6.Hsu CC, Wang H, Hsu YH, Chuang SY, Huang YW, Chang YK, Liu JS, Hsiung CA, Tsai HJ. Use of Nonsteroidal Anti-Inflammatory Drugs and Risk of Chronic Kidney Disease in Subjects With Hypertension: Nationwide Longitudinal Cohort Study. Hypertension. 2015; 66(3): 524-33. doi: 10.1161/HYPERTENSIONAHA. 114.05105.

7.Kuo KL, Hung SC, Liu JS, Chang YK, Hsu CC, Tarng C. Iron supplementation associates with low mortality in pre-dialyzed advanced chronic kidney disease patients receiving erythropoiesis-stimulating agents: a nationwide database analysis. Nephrol Dial Transplant. 2015; 30(9): 1518-25. doi: 10.1093/ndt/gfv085.

8.Chang YK, Hsu CC, Hwang SJ, Chen PC, Huang CC, Li TC, Sung FC. A Comparative Assessment on Survival between Propensity Score Matched Patients with Peritoneal Dialysis and Hemodialysis in Taiwan. Medicine. 2012; 91(3): 144-151. doi: 10.1097/MD.0b013e318256538e.

9.Nguyen TL, Collins GS, Spence J, Daures JP, Devereaux PJ, Landais P, et al. Double-adjustment in propensity score matching analysis: choosing a threshold for considering residual imbalance. BMC Med Res Methodol. 2017;17(1):78. Epub 2017/04/30. doi: 10.1186/s12874-017-0338-0. PubMed PMID: 28454568; PubMed Central PMCID: PMCPMC5408373.

---

## [Decision Letter · Decision Letter 2]

15 Aug 2022

Association Between Atrial Fibrillation and Risk of End-Stage Renal Disease Among Adults With Diabetes Mellitus

PONE-D-21-17026R2

Dear Dr. Lim,

We’re pleased to inform you that your manuscript has been judged scientifically suitable for publication and will be formally accepted for publication once it meets all outstanding technical requirements.

Kind regards,

Abduzhappar Gaipov

Academic Editor

PLOS ONE

Additional Editor Comments (optional):

Reviewers' comments:

Reviewer's Responses to Questions

**Comments to the Author**

1. If the authors have adequately addressed your comments raised in a previous round of review and you feel that this manuscript is now acceptable for publication, you may indicate that here to bypass the “Comments to the Author” section, enter your conflict of interest statement in the “Confidential to Editor” section, and submit your "Accept" recommendation.

Reviewer #1: All comments have been addressed

Reviewer #2: (No Response)

2. Is the manuscript technically sound, and do the data support the conclusions?

Reviewer #1: Yes

Reviewer #2: Partly

3. Has the statistical analysis been performed appropriately and rigorously? 

Reviewer #1: Yes

Reviewer #2: No

4. Have the authors made all data underlying the findings in their manuscript fully available?

Reviewer #1: (No Response)

Reviewer #2: No

5. Is the manuscript presented in an intelligible fashion and written in standard English?

Reviewer #1: Yes

Reviewer #2: Yes

6. Review Comments to the Author

Reviewer #1: There is no further comments. All comment were addressed by the authors.

Reviewer #2: I do not fully agree with the authors' responses and selected analytical plan, but theoretically the results should be quite similar (to other alternative proposed approach)

7. PLOS authors have the option to publish the peer review history of their article (what does this mean?). If published, this will include your full peer review and any attached files.

Reviewer #1: No

Reviewer #2: No

---

## [Editor Report · Acceptance letter]

17 Aug 2022

PONE-D-21-17026R2 

Association Between Atrial Fibrillation and Risk of End-Stage Renal Disease Among Adults With Diabetes Mellitus 

Dear Dr. Lim:

I'm pleased to inform you that your manuscript has been deemed suitable for publication in PLOS ONE. Congratulations! Your manuscript is now with our production department. 

Kind regards, 

on behalf of

Dr. Abduzhappar Gaipov 

Academic Editor

PLOS ONE